# Metabolic Risk Factors and Survival in Patients with Glioblastoma

**DOI:** 10.3390/cancers16213666

**Published:** 2024-10-30

**Authors:** John Paul Aboubechara, Orwa Aboud

**Affiliations:** 1Department of Neurology, University of California, Davis, Sacramento, CA 95817, USA; oaboud@ucdavis.edu; 2UC Davis Comprehensive Cancer Center, University of California, Davis, Sacramento, CA 95817, USA; 3Department of Neurological Surgery, University of California, Davis, Sacramento, CA 95817, USA

**Keywords:** glioblastoma, metabolic syndrome, hyperglycemia, dyslipidemia, overall survival

## Abstract

Metabolic syndrome has been associated with systemic cancers but its association with patients with glioblastoma is unclear. This study retrospectively examined patients with IDH wild-type glioblastoma and demonstrated that metabolic syndrome is more prevalent in glioblastoma patients (41%) compared to the general population (33%). However, after adjusting for confounders, metabolic syndrome is not significantly linked to overall survival (*p* = 0.1). Nonetheless, the accumulation of metabolic risk factors correlates with decreased survival (*p* = 0.03), with hyperglycemia identified as a significant independent risk factor (*p* = 0.05). These findings suggest that while metabolic syndrome may not affect survival significantly, hyperglycemia does. This highlights the need for further research and targeted clinical management.

## 1. Introduction

A hallmark of cancer is altered cellular metabolism, which allows cancer cells to meet the high demands of rapid proliferation and survival in often hostile environments [1]. One of the earliest and most well-known observations of this metabolic reprogramming is described as the Warburg effect [2,3]. In this phenomenon, rapidly dividing cancer cells predominantly convert glucose to lactate through aerobic glycolysis, even in the presence of sufficient oxygen, which is a less efficient way of generating ATP compared to oxidative phosphorylation [2]. However, this metabolic shift provides cancer cells with several advantages, including the diversion of metabolic intermediates for anabolic processes that support the biosynthesis of macromolecules necessary for cell growth and division [4].

For gliomas, the identification of isocitrate dehydrogenase (IDH) mutations as a critical marker of tumor behavior and prognosis has invigorated efforts to better understand glioma metabolism [5]. IDH is a central enzyme in the tricarboxylic acid cycle that underlies much of cellular energy production [5]. Mutations of this enzyme have been shown to alter cellular metabolism, which is thought to drive glioma cell growth [5]. Outside of the nervous system, systemic metabolic dysregulation manifesting as metabolic syndrome has been associated with an increased prevalence of systemic cancers, with the risk being proportional to the number of metabolic risk factors experienced by a patient [6,7]. Mechanisms underlying this association remain under investigation and include the production of growth factors, cytokines, and other signaling molecules that drive tumor growth [8]. An example of such a mechanism includes the effects of insulin resistance in increasing the levels of insulin-like growth factor 1, which stimulates cancer cell proliferation and angiogenesis [8].

Recent studies have explored this association with gliomas but have had conflicting results [9,10,11]. One study demonstrated a significant association between metabolic syndrome and survival [10]; another study demonstrated a trend toward worsened survival [9]; while a third study actually found the opposite trend—improved overall survival in patients with metabolic syndrome [11]. These contradictory findings may stem from the methodology of the studies and the patient populations. Thus, this study aims to expand on these prior results in testing the hypothesis that patients with pre-existing metabolic syndrome have an increased prevalence of IDH wild-type glioblastoma and worse survival outcomes. This study emphasizes the examination of patients solely with IDH wild-type glioblastoma as IDH-mutant gliomas are known to have different metabolic features [12].

## 2. Materials and Methods

A retrospective cohort study was conducted examining 73 patients with pathologically confirmed glioblastoma isocitrate dehydrogenase (IDH)-wild-type treated at the University of California, Davis, between 2018 and 2023. This study was conducted in accordance with the Declaration of Helsinki and it was given an exempt status by the Institutional Review Board of the University of California, Davis, 95,817, on 11 April 2023 due to this being a retrospective review of deidentified data available in the electronic medical record. All patients were adults, and they underwent surgical resection followed by standard of care chemoradiation [13]—though some were unable to complete treatment due to poor functional status or decision to transition to hospice care. Electronic medical records were reviewed to identify whether these patients had developed metabolic syndrome prior to diagnosis of the tumor. Records reviewed included progression-free survival, overall survival, treatment, pathologic diagnosis, tumor molecular characteristics, blood pressure, body mass index, medications, and laboratory data.

Metabolic syndrome was defined based on the internationally agreed-upon definition [14]. Based on this definition, patients meet the criteria for metabolic syndrome if they have three of the five following criteria: obesity, hypertension, hyperglycemia, decreased high-density lipoprotein (HDL), and hypertriglyceridemia. Obesity is defined as a body mass index (BMI) greater than 30 kg/m^2^. Hypertension is defined as systolic blood pressure greater than 130, diastolic blood pressure greater than 90, or currently taking antihypertensives. Hyperglycemia is determined by fasting blood glucose greater than 100, HgA1c greater than 6.0, or the patient currently taking antihyperglycemics. Decreased HDL must be less than 40 mg/dL in males or less than 50 mg/dL in females; it is also defined as a patient currently taking a statin medication. Hypertriglyceridemia requires a triglyceride level greater than 150 mg/dL. It is important to emphasize that these features were captured prior to tumor diagnosis and treatment since certain treatments can exacerbate these metabolic risk factors (e.g., steroid use).

For calculation of overall survival, the date of diagnosis was defined as the date of the first surgery that led to tumor diagnosis. The date of death was determined as either the confirmed death date or the final electronic medical record (EMR) entry after being placed in hospice care. All patients in this study died from this malignancy. Statistical analyses included unpaired *t*-tests when comparing two groups and linear regression models for comparison of two continuous variables. Kaplan–Meier curves were used to depict overall survival between groups over time. Statistical significance was drawn for *p*-values equal to or less than 0.05.

## 3. Results

### 3.1. Patient Characteristics

Table 1 depicts the characteristics of the patients involved in this study. All seventy-three patients had IDH wild-type glioblastoma. Metabolic syndrome was identified in 41% of the patients (30/73) (Table 1), which is greater than the estimated prevalence of metabolic syndrome of 33% in the general adult population in the United States [15]. However, the mean age of the patients in our cohort was 65.2 [95% CI; 62.40, 68.09], which is greater than the mean age of the general population, as reported by Hirode et al., wherein the majority of their patients were aged less than 60—though no overall quantification is reported [15]. Patients with metabolic syndrome were significantly older than those without (69.7 vs. 62.1 years, *p* = 0.01) (Appendix A). Thirty patients (41%) met the above criteria for metabolic syndrome, of which 10 (14%) were female and 20 (27%) were male. Among those without metabolic syndrome, 15 (21%) were female and 28 (38%) were male. There was a similar distribution of race between the two groups, with the White race making up 77% of those with metabolic syndrome and 65% of those without. Those without metabolic syndrome tended to have a higher prevalence of gross total resection, methylguanine methyltransferase (MGMT) methylation, and epidermal growth factor receptor (EGFR) amplification (Table 1).

### 3.2. Metabolic Syndrome and Individual Risk Factors Are Associated with Worsened Overall Survival

Overall survival was defined from the date of the first surgery to the date of death. Patients with metabolic syndrome had a significant reduction in mean overall survival (7.9 vs. 15.0 months, *p*-value 0.01) prior to correcting for confounding variables (Figure 1B). A Kaplan–Meier plot also demonstrates this result and demonstrates a reduction in median overall survival for patients with metabolic syndrome (5.8 vs. 11.4 months) (Figure 1A). Evaluation for confounding variables demonstrated that patients with metabolic syndrome were older on average (Table 1). Age also demonstrated a significant negative association with overall survival as discussed below. After correction for age differences between patients with and without metabolic syndrome, metabolic syndrome demonstrated a trend for worsened survival (7.9 vs. 12.5 months, *p* = 0.1) (Figure 1C).

Given this trend, we conjectured that rather than evaluating survival simply based on the criteria of metabolic syndrome, it may be valuable to evaluate metabolic risk factors individually and as patients accumulate additional risk factors. As patients accumulate additional risk factors, there is a significant negative association with overall survival (slope = −2.5 months per additional risk factor, *p* = 0.03) (Figure 1D). However, the largest change in survival resulted from the accumulation of the first risk factor, while subsequent risk factors had less of an effect on survival.

We next examined each metabolic risk factor individually as it affects overall survival in our patients. Given that we previously identified a strong negative association between age and overall survival, all subsequent analyses include a correction factor for age differences. Despite correction for age differences, hyperglycemia demonstrated a significant negative association with overall survival (11.2 vs. 16.9 years, *p* = 0.05) (Figure 2C). Hypertension demonstrated a trend for worsened overall survival (11.9 vs. 16.8 years, *p* = 0.1) (Figure 2B). Obesity, decreased HDL, and hypertriglyceridemia did not affect overall survival (Figure 2A,D,E).

Analyses were performed to examine whether the following patient characteristics affected overall survival: age, race, sex, resection status, MGMT methylation status, and EGFR amplification status. As discussed previously, age demonstrated a strong negative association with overall survival (slope = −0.33 months of survival per year of age, *p* = 0.004) (Figure 3A). Race and sex did not lead to a significant effect on overall survival (Figure 3B,C). Resection status did demonstrate a positive association with survival with gross total resection (GTR) having greater overall survival compared to biopsy (14.8 months vs. 5.7 months, *p* = 0.05) (Figure 4A). MGMT methylation was also associated with improved survival compared to unmethylated status (15.6 months vs. 9.8 months, *p* = 0.04) (Figure 4B). EGFR amplification was not associated with survival (12.8 months vs. 9.1 months, *p* = 0.29) (Figure 4C).

## 4. Discussion

Metabolic syndrome has been associated with increased prevalence and decreased survival in multiple types of cancer, including breast, liver, colorectal, bladder, pancreatic, and endometrial [6], though a mechanistic understanding is lacking. One hypothesized mechanism suggests that metabolic dysregulation leads to a systemic inflammatory state in which cytokines and growth factors can facilitate cancer cell proliferation [16]. Another proposed mechanism is that metabolic syndrome leads to insulin resistance and hyperinsulinemia, which leads to increased insulin-like growth factor (IGF 1) activity, which may lead to the development and progression of tumors [17].

### 4.1. Association Between the Prevalence of Metabolic Syndrome and the Development of Glioblastoma

On the other hand, the association between metabolic syndrome and glioblastoma pathogenesis has not been sufficiently explored. Three prior studies have examined this association with conflicting results. Rogers et al. examined a cohort of patients from Cleveland, Ohio, and demonstrated a slightly higher prevalence of metabolic syndrome among patients who developed glioblastoma compared to the general population (35.6% vs. 34.7%, respectively) [9]. McManus et al. similarly examined this association in a cohort of patients from New Zealand and also showed a slight increase in the prevalence of metabolic syndrome among patients who developed glioblastoma (18.2% vs. 16%) [10]. Lucas et al. examined a cohort of patients from Portugal and demonstrated the opposite association, having observed a much lower prevalence of metabolic syndrome among patients who developed glioblastoma (11.1% vs. 32.7%) [11]. Our results demonstrate a higher prevalence of metabolic syndrome in patients who developed glioblastoma (41% vs. 33%) (Table 1). This result, however, is confounded by the fact that the mean age of the patients in our cohort is greater than that of Hirode et al.’s study, which examined the prevalence of metabolic syndrome in the general adult population [15]. That study went on to demonstrate that the prevalence of metabolic syndrome was greater with increasing age. As such, it is possible that the increased prevalence of metabolic syndrome in our patients compared to the general population is due to older age. These findings are interesting but are limited to retrospective studies, and as such, future prospective studies are needed to better evaluate this association. If the association is validated, then metabolic risk factor modification could prove to be a promising approach to the treatment of patients with glioblastoma. Further, studies examining the mechanisms underlying this association could identify future targets for drug development.

### 4.2. Association Between Metabolic Syndrome and Overall Survival in Patients with Glioblastoma

Two of the previously referenced studies demonstrated that metabolic syndrome was associated with worsened survival. McManus et al. demonstrated a significantly decreased overall survival in patients with metabolic syndrome (8.0 months vs. 13.0 months, *p* = 0.016) [10]. Rogers et al. demonstrated only a trend towards worsened survival in patients with metabolic syndrome (7.7 months vs. 12.7 months, *p* = 0.22); however, they identified a significant association of metabolic syndrome with decreased survival in a subset of patients who completed the full schedule of concurrent chemoradiation (12.4 months vs. 17.9 months, *p* = 0.18). Lucas et al., on the other hand, demonstrated a trend towards improved survival in patients with metabolic syndrome who later developed glioblastoma (19.8 months vs. 17.7 months, *p* = 0.085) [11]. Our results in Figure 1 demonstrate that metabolic syndrome in patients who later developed glioblastoma is associated with a trend toward decreased survival (7.9 vs. 12.5 months, *p* = 0.1)—after correction for the confounding variable of age, which itself is a significant prognostic factor. Interestingly, there was a significant association between the accumulation of additional metabolic risk factors and decreased overall survival (Figure 1D). To our knowledge, this is the first study to demonstrate this somewhat linear association between the number of metabolic risk factors a patient has and overall survival.

Metabolic syndrome, encompassing insulin resistance, hyperglycemia, obesity, dyslipidemia, and hypertension, significantly affects glioblastoma prognosis, with chronic inflammation proposed to play a central convergent role [8]. Inflammatory processes associated with obesity and metabolic syndrome, including the release of pro-inflammatory cytokines like TNF-α and IL-6, create a tumor-promoting environment that enhances cancer cell proliferation, invasion, and resistance to apoptosis [8]. This chronic low-grade inflammation contributes to an aggressive glioblastoma phenotype by activating pathways such as NF-κB and STAT3, which drive tumor progression and survival [8,18]. The chronic inflammatory state, alongside metabolic imbalances, not only accelerates tumor progression but also diminishes the effectiveness of standard treatments, contributing to poorer survival outcomes in glioblastoma patients with metabolic syndrome [19]. Addressing these inflammatory and metabolic factors may be crucial in improving long-term treatment efficacy and survival.

### 4.3. Association of Hyperglycemia with Worsened Overall Survival

In our analysis, hyperglycemia emerged as an independent risk factor associated with decreased overall survival (11.2 vs. 16.9 years, *p* = 0.05) (Figure 2), while hypertension demonstrated a trend toward worsened survival. Hyperglycemia in patients prior to the diagnosis of glioblastoma has been demonstrated by multiple studies to be associated with decreased overall survival [20,21,22]. Multiple hypotheses have been proposed to explain why hyperglycemia may worsen survival.

Glioblastomas are highly dependent on glycolysis for energy production, even in the presence of oxygen—a phenomenon known as the Warburg effect. This metabolic reprogramming allows cancer cells to bypass oxidative phosphorylation to convert glucose into energy [4]. Although this is an inefficient mechanism for energy generation, this allows the diversion of metabolites of the tricarboxylic acid cycle into anabolic processes to facilitate cell growth and division [4]. This reliance on glucose metabolism through glycolysis for energy production and cell proliferation makes gliomas particularly vulnerable to changes in glucose levels, creating a potential target for therapeutic intervention [4,23].

Another contributing factor is insulin resistance, which is often associated with hyperinsulinemia—a condition where the body compensates for impaired insulin function by producing excess insulin [24]. Elevated insulin levels can enhance tumor growth through the insulin-like growth factor 1 (IGF-1) pathway, a signaling cascade known to promote cellular proliferation and inhibit apoptosis (programmed cell death) [20,21,22]. As IGF-1 receptors are commonly overexpressed in glioblastoma cells, this pathway represents a key mechanism by which metabolic disturbances can exacerbate tumor progression [20,21,22].

The association between hyperglycemia and worsened overall survival in glioblastoma patients introduces a clinical challenge, particularly regarding the use of corticosteroids. Corticosteroids are a cornerstone in the management of cerebral edema related to brain tumors and post-operative recovery [25]. However, they are also known to induce hyperglycemia, potentially aggravating the metabolic environment and facilitating tumor growth [25]. This paradox highlights a difficult balance between managing the immediate complications of brain swelling and addressing the long-term implications of hyperglycemia in tumor biology [25].

Given these complexities, several studies have begun investigating the potential of targeting hyperglycemia and glucose metabolism in glioblastoma treatment. Approaches such as ketogenic diets, glucose-lowering medications, and metabolic inhibitors are being explored to limit glucose availability to the tumor [26,27,28,29]. While preclinical models have shown some promise, clinical trials have yielded limited success thus far, with inconsistent results and challenges in translating metabolic therapies into significant survival benefits for patients [26,27,28,29]. This area of research remains ongoing, with the hope that future studies may offer more effective strategies for targeting the metabolic vulnerabilities of glioblastoma.

### 4.4. Strengths and Limitations

The prior studies included both IDH wild-type and IDH mutant tumors in their analyses, though the proportion of patients with IDH mutant tumors was generally small and likely not to have affected their results considerably [9,10,11]. As IDH mutant tumors have been shown to undergo different metabolisms [12], this study focused solely on IDH wild-type tumors, which improves the generalizability of our findings. This study also demonstrated that the extent of resection and MGMT promoter methylation status were important prognostic factors that were significantly associated with overall survival. These results also validate the generalizability of our dataset by re-demonstrating these associations with overall survival [30,31].

There are limitations to the results presented in this study. This study had a limited sample size, leading to relatively low power of the statistical results. Further, as a retrospective study, the results presented here are associations from which causality should not be assumed. Additionally, the retrospective nature of this study meant that there were group differences for which corrections had to be made in order to make broader inferences. For example, patients with metabolic syndrome were significantly older and age was significantly associated with worsened survival; thus, correction for age had to be performed to examine the effect of metabolic syndrome on survival. Future prospective clinical studies and further mechanistic research are critical to clarify whether there is a meaningful association between metabolic syndrome and survival in patients who develop glioblastoma.

## 5. Conclusions

Metabolic syndrome has been associated with tumorigenesis and decreased survival in multiple cancer types, though its association with glioma pathogenesis remains unclear. This study demonstrates that metabolic syndrome may have a higher prevalence in patients with glioblastoma compared to the general population, and metabolic risk factors are associated with decreased overall survival. Hyperglycemia appears to be the strongest driver of this finding, which has been supported by multiple prior studies. Additional prospective clinical studies and mechanistic research are needed to better evaluate these associations. If confirmed, clinical strategies to target metabolic risk factors may play an important role in the treatment of patients with glioblastoma.

## Figures and Tables

**Figure 1 cancers-16-03666-f001:**
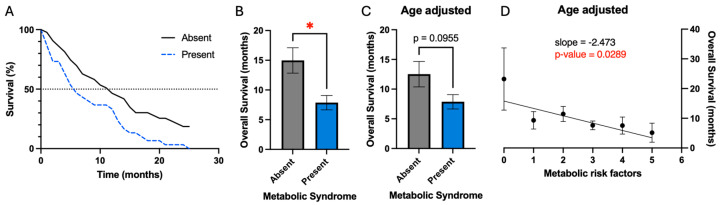
Metabolic risk factors are associated with worsened overall survival. (**A**) Kaplan–Meier plot demonstrating worsened overall survival in patients with metabolic syndrome. (**B**) Patients with metabolic syndrome have decreased mean overall survival; * signifies *p*-value ≤ 0.05. (**C**) After correction for age as a confounding variable, metabolic syndrome only demonstrates a trend of worsened survival. (**D**) Accumulation of increasing numbers of metabolic risk factors is associated with worsened survival.

**Figure 2 cancers-16-03666-f002:**
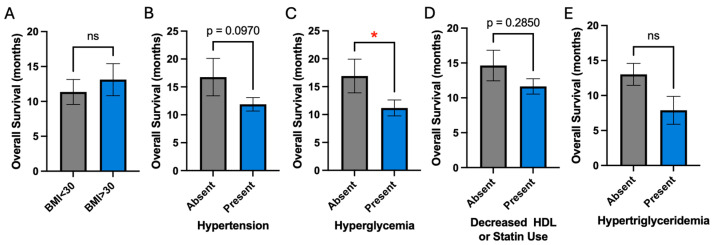
Association of individual metabolic risk factors with overall survival. (**A**,**B**,**D**,**E**) BMI, hypertension, decreased HDL or statin use, and hypertriglyceridemia are not associated with survival. (**C**) Hyperglycemia is significantly associated with worsened survival; * signifies *p*-value ≤ 0.05; ‘ns’ denotes not significant.

**Figure 3 cancers-16-03666-f003:**
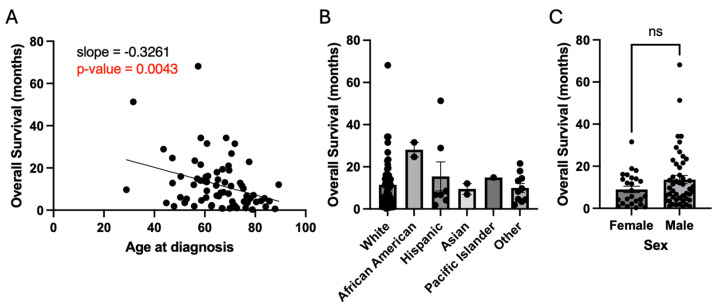
Patient demographics and overall survival. (**A**) Older age at diagnosis has a directly proportional association with worsened survival. (**B**,**C**) Race and sex are not associated with survival; ‘ns’ denotes not significant.

**Figure 4 cancers-16-03666-f004:**
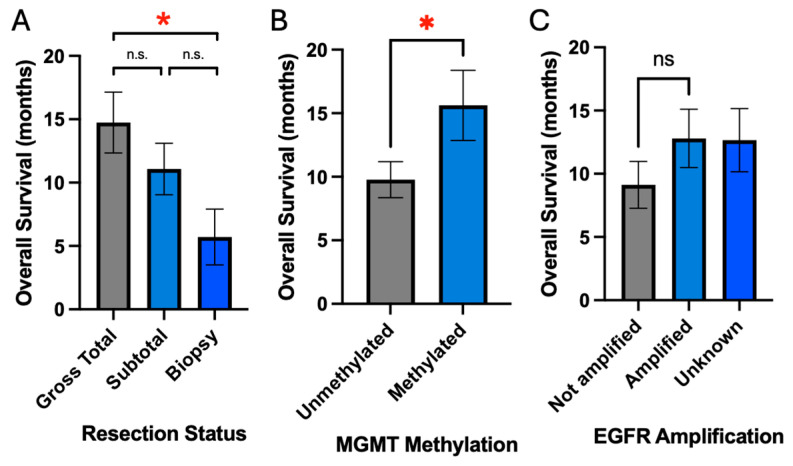
Extent of resection and tumor molecular markers are associated with overall survival. (**A**) Increased extent of resection is associated with improved survival. (**B**,**C**) MGMT methylation but not EGFR amplification is associated with improved survival; * signifies *p*-value ≤ 0.05, while ‘ns’ denotes not significant.

**Table 1 cancers-16-03666-t001:** Patient Characteristics. Comparison of the demographics, treatment status, and tumor molecular features between patients with and without metabolic syndrome.

	MetS ^1^ Absent	MetS Present
Total (Percent)	43 (59%)	30 (41%)
Female	15	10
Male	28	20
White	28	23
Age, mean (95% C.I.)	62.1 (58.3–66.0)	69.7 (65.8–73.6)
Hispanic	6	1
African American	1	1
Asian	1	1
Pacific Islander	1	0
Other/Unknown	6	4
Gross total resection	23	10
Subtotal resection	13	15
Biopsy	5	5
Unknown resection	2	0
MGMT methylated	19	10
MGMT not methylated	23	20
EGFR amplified	21	16
EGFR not amplified	8	12
EGFR Unknown	14	2

^1^ MetS: metabolic syndrome.

## Data Availability

All data can be made available upon request.

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
