# Peer review of "Metabolic Risk Factors and Survival in Patients with Glioblastoma"

_cancers, 2024, doi:10.3390/cancers16213666_

Round 1
Reviewer 1 Report
Comments and Suggestions for Authors
Thank you for considering me as a reviewer for this highly interesting manuscript titled "Metabolic Risk Factors and Survival in Patients with Glioblastoma".
In the manuscript, the authors present a comparison of metabolic syndrom rates in a case-control study of glioblastoma patients with and without meatbolic syndrom. The manuscript is well written, addresses a highly relevant topic and applies a sound methodological approach as far as the analyses comparing glioblastoma patients with and without metabolic syndrom are concerned.
My major concern with the manuscript is that in the discussion as well as conclusion section and abstract the finding that metabolic syndrome is more common than in the general population. Considering this, the methodological approach for the comparison with the general population needs to be described more clearly and to support the drawn conclusions. If nothing else, the likely different age distributions in both cohorts need to be considered since it's quite plausible that the mean age in the glioblastoma cohort was significantly higher than in the general population. Could the observed higher rate of metabolic syndrome in the glioblastoma cohort not at least to a relevant extent be attributable to a the higher age as obvious risk factor for the metabolic syndrome? To support the conclusions presented in regard to the rate of metabolic syndrome compared to the general population, an at least age-adjusted comparison seems mandatory to me.
Author Response
Please find response to Reviewer 1 in the attached document.

Reviewer 2 Report
Comments and Suggestions for Authors
The research article titled, "Metabolic risk factors and survival in patients with Glioblastoma" is a significant addition to this field. The research question addressed here is novel and needs to be investigated. The article is written clearly for the readers to understand easily.
A few comments/ questions and suggestions to the authors.
1)In the methods section, authors mentioned that statistical significance was drawn for p-values less than 0.05 and it followed and accepted globally. However, when p-value was equal to 0.05, authors concluded that there is significant association. For eg: lines 130-131; lines 144-146. Is there any reason for this conclusion when p-value = 0.05?
2) Did the authors do a multivariate analysis using Cox-regression model?
3) The results show only the statistical tests with metabolic syndrome and survival mainly. Did the authors look at the association of other clinical factors like stage, grade, treatment (chemoradiation) with survival?
4) How many patients attended the hospital with Glioblastoma during the prescribed period and how many (%) patients had IDH-wild type GBM?
5)Did the treatment (chemo or radiation) has any effect on the metabolic syndrome? How were the metabolic factors before and after chemoradiation?
6) Authors mentioned that the sample size is low. Do they intend to extend this study to a larger cohort of GBM patients?
7) If possible, try to include more references from 2021-2024. Please include references for lines 40 (altered cellular metabolism as a hallmark) and 41 (Warburg effect).
Author Response
Please find response to Reviewer 2 in the attached document.
